# Climate Chamber Experiment Study on the Association of Turning off Air Conditioning with Human Thermal Sensation and Skin Temperature

**Yiwen Jian [1,2,\*], Shuwei Liu [1], Mengmeng Bian [3], Zijia Liu [3] and Shengjie Liu [1]**

[1] Faculty of Urban Construction, Beijing University of Technology, Beijing 100124, China; liusw0214@163.com (S.L.); liushengjie1003@126.com (S.L.)
[2] Beijing Key Laboratory of Green Built Environment and Energy Efficient Technology, Beijing 100124, China
[3] State Key Laboratory of Building Safety and Built Environment, China Academy of Building Research, Beijing 100013, China; bianmengm@126.com (M.B.); liuzijia@chinaibee.com (Z.L.)
\* Correspondence: jianyiwen@bjut.edu.cn; Tel.: +86-131-2659-2972

**Abstract:** To date, few attempts have been made to associate air conditioning behavior with environmental conditions and the occupants' thermal sensations and physiological states simultaneously. In this study, a series of experiments were conducted in a climate chamber environment, representative of a typical intermittent air conditioning process in residences. For 29 participants, local skin temperatures, thermal sensation and the participants' intention of turning off air conditioning were recorded continually. Skin temperature and thermal sensation were found to keep decreasing over time, which in turn triggered turning off air conditioning. It is also noted that participants reported different thermal sensations when they intended to turn off air conditioning. However, there was no statistically significant difference in skin temperature of exposed body parts such as foot and calf. Additionally, given the ambient set temperature, the probability of turning off air conditioning exponentially increased with the increasing air conditioning duration. Accordingly, from a physiological perspective, the occupants' behavior of turning off air conditioning was largely dependent on the local skin temperature of exposed body parts. From an environmental perspective, air conditioning duration demonstrated influences on the air conditioner switch-off. The lower the ambient temperature or the longer the exposure to air conditioning environment, the stronger the intention of the participants to turn off air conditioning.

**Keywords:** turning off air conditioning; skin temperature; thermal sensation; physiological; environmental

## 1. Introduction

Buildings consume approximately one-third of the total primary energy resources [1]. The occupants' behaviors such as switching on/off blinds, opening/closing windows, turning on/off fans and air conditioning, adjusting the thermostat, and dimming lights have great influence on buildings' energy consumption [2–7]. The energy consumption of air conditioning in Chinese residential buildings has increased substantially due to the acceleration of urbanization and the increasing use of air conditioning in recent years [8].

The occupants' behavior related to turning on/off air conditioning in buildings not only affects the thermal comfort, but also impacts energy consumption. The temperature set point was identified as the most influential environmental parameter on thermal comfort and energy consumption, compared with the use of blinds, lighting systems, windows, and fans [3]. Furthermore, the occupants' behavior related to air conditioning systems can result in significant differences in building energy consumption [9–11]. When the occupants' behavior is not properly taken into consideration in simulation models, this can result in overestimating or underestimating building energy consumptions. Thus, understanding the occupants' behavior related to air conditioning systems is important in

reducing building energy consumption for space cooling and enhancing indoor thermal comfort. In Chinese residential buildings, split air conditioners are the most commonly used and run intermittently to meet the temperature set point based on the occupants' thermal preferences. Once the air conditioner is turned on, the room temperature immediately decreases in the first few minutes and then remains stable. Ignoring the influence of factors such as income level, awareness of energy conservation, and the electricity prices, the feelings of warm and cool are the main factors that affect the air conditioning behavior of the occupants.

Thermal sensation, related to both the physiological state of the human body and environmental conditions, is one of the most important indexes for evaluating the feeling of warm or cool in thermal comfort studies. Research found that skin temperature under environmental stimuli performs an important role in thermal sensation. Wang et al. [12] and Zhang et al. [13] found skin temperature to be an index to indicate thermal sensation under a cooling environment. Chaudhuri et al. [14] suggested that the skin temperature of hands had strong correlation with thermal sensation. Wang et al. [15] found that people have a warm sensation when the temperature of the finger is 2 °C above the forearm. Several studies developed predictive models of thermal sensation vote (TSV) as a function of mean skin temperature, local skin temperature, and skin temperature change rate [16–19]. The studies were usually conducted in the environments such as the transitional environment between indoor and outdoor [13,15,20–23], gradient transitional environments [14,19,24], and stable environments [25–27]. Those existing experimental scenarios are not suitable for representing the air conditioning process in which the split air conditioners start and run intermittently in Chinese residential buildings.

Environmental conditions, the physiological state of the human body, thermal sensation, and air conditioning behavior are interconnected, and the change in thermal sensation and air conditioning behavior is triggered by the changes in the physiological state of the human body, which ultimately results from the ambient thermal environment. From the environment-driven perspective, most research in the field of air conditioning behavior has focused on the driving forces of the occupants' behavior related to turning on/off air conditioning and regression models relating the probability of turning on/off air conditioners to indoor temperature [11,28,29] or outdoor temperature [30–32]. Furthermore, some studies analyzed how physiological state and thermal sensation affect the occupants' behavior of turning on/off air conditioning for thermal adaptation, and noted the important role of human skin temperature. For example, Schlader et al. [33,34] and Song et al. [35] found that skin temperature appeared to be a vital parameter influencing the occupants' behavior. Vargas [36] stated that the occupants' behavior was dependent on the magnitude of increases in mean skin temperature, and Jacquot et al. [19] noted that wrist temperature was important for the prediction of the occupants' behavior. The above studies were based on an experiment conducted in a natural environment of a multi-person classroom [35] or were conducted in an environmental chamber with high/low temperature over 30 °C or below 10 °C [33,34,36]. Few attempts have been made to associate air conditioning behavior in summer with the occupants' physiological state and their thermal sensation for an intermittent air conditioning environment that is commonly found in Chinese residential buildings.

Thus, in this study, a series of experiments were performed in an environmental chamber by monitoring skin temperature, TSV, and the patterns of air conditioning duration under typical dynamic cooling scenarios in Chinese residential buildings, aiming to reach the following objectives:

(1) Analyze the relationship between the occupants' behavior of turning off air conditioning and physiological state (e.g., mean skin temperature, local skin temperature) as well as thermal sensation;

(2) Understand the driving forces of the occupants' behavior of turning off air conditioning from both physiological and environmental perspectives.

## 2. Methods

### 2.1. Participants

A total of 29 healthy participants (13 males and 16 females) were recruited to participate in all the experimental scenarios of this study. All participants were young students with ages ranging from 21 to 23 years of age and they lived in Beijing for more than one year. The characteristics of the participants are listed in Table 1. All participants were in normal physical condition, emotionally stable, and they were required to avoid doing strenuous exercise before the experiment. In addition, all participants were asked not to have food, tea, or energy drinks within one hour before the experiment. They were instructed to wear short sleeves, short pants, and slippers so that the clothing style of the participants was consistent with that of occupants in Chinese residential buildings during summer. The clothing insulation value of the participants was 0.39 clo.

**Table 1.** Demographic information of participants.

| Variables | Mean | St. dev. | Max. | Min. |
|---|---|---|---|---|
| Height (cm) | 169 | 13 | 187 | 155 |
| Weight (kg) | 60.7 | 0.1 | 82 | 44.5 |
| Age | 22.1 | 3 | 23 | 21 |
| Living time in Beijing (year) | 3.3 | 2.8 | 4 | 2 |

### 2.2. Experimental Scenarios

The purpose of this study was to investigate the occupants' physiological thermal response and behavior of turning off air conditioning in residential buildings in summer. For this, the scenarios should be chosen to represent the typical air conditioning environment in Chinese residential buildings. Air conditioning in these buildings were frequently switched on when the ambient temperature was approximately 29 °C [37–39]. Furthermore, the results from survey questionnaires indicated that the air conditioning temperature was usually set at 27 °C, 25 °C, or 23 °C [40]. Therefore, three experimental scenarios (A: 29→27 °C, B: 29→25 °C, and C: 29→23 °C) were chosen in this study as shown in Figure 1a–c. In addition, the experimental scenario (D: 27→23 °C) represented in Figure 1d was also chosen to explore the influence of initial ambient temperature on the occupants' behavior of turning off air conditioning when the set point temperature for air conditioning was the same. As shown in Figure 1, for scenario A (29→27 °C), B (29→25 °C), and C (29→23 °C), the ambient temperature was first set at 29 °C for 30 min; then ambient temperature decreased to 27 °C, 25 °C, and 23 °C, respectively, during the next 30 min; after that, the ambient temperature was stabilized in the last 60 min. Similarly, in scenario D (27→23 °C), the ambient temperature was set at 27 °C initially for 30 min, and was finally stabilized at 23 °C in the last 60 min.

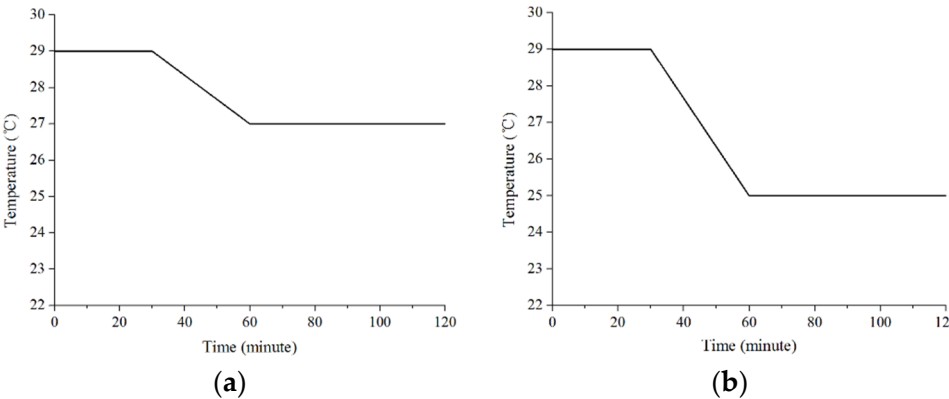

**Figure 1.** *Cont.*

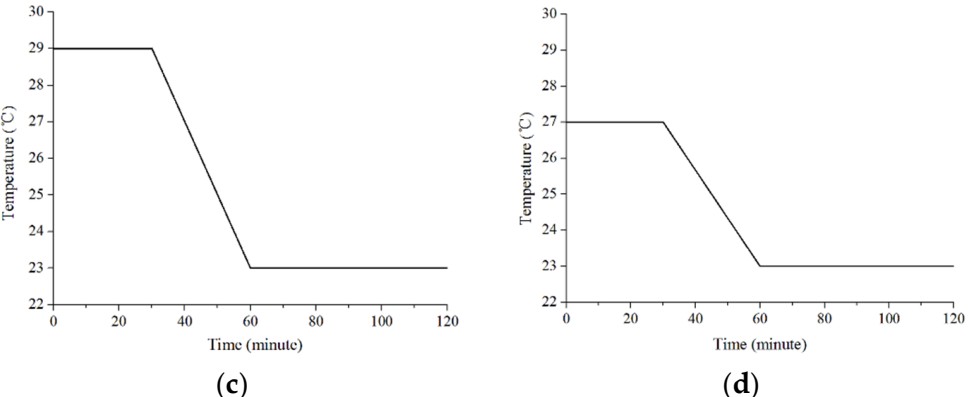

**Figure 1.** Timelines of the four experimental scenarios. (**a**) Scenario A: 29→27 °C. (**b**) Scenario B: 29→25 °C. (**c**) Scenario C: 29→23 °C. (**d**) Scenario D: 27→23 °C.

### 2.3. Measurements

#### 2.3.1. Environmental Chamber Measurements

The experiments in this study were conducted in an environmental chamber in Beijing. The indoor environment in the chamber was controlled by an overhead mechanical cooling system. This all-air system supplies and returns air through the orifice panels at the top of the room and the return air outlets at the side wall location, respectively. During the experiment, indoor environment parameters including dry bulb temperature, relative humidity, wet-bulb globe temperature (WBGT) and air velocity were monitored every minute using a thermal environment analysis instrument (HD32.3, DeltaOHM, Padua, Italy) placed at the height of 1.2 m. Detailed information of the thermal environment analysis instrument is summarized in Table 2. During the experiments, the air velocity in the chamber was controlled at less than 0.2 m/s according to the ASHRAE 55–2017 requirements [41].

**Table 2.** Information of data acquisition devices.

| Parameter | Accuracy | Resolution | Instrument Model |
|---|---|---|---|
| Dry bulb temperature | ±0.3 °C | 0.1 °C | HD32.3 |
| Relative humidity | ±2% | 0.1% | HD32.3 |
| WBGT | ±0.3 °C | 0.1 °C | HD32.3 |
| Air velocity | ±0.03 m/s | 0.01 m/s | HD32.3 |
| Skin temperature | ±0.5 °C | 0.0625 °C | DS1922L |

#### 2.3.2. Skin Temperature Measurements

The mean skin temperature can be better reflected if the skin temperature measurement points are over seven [42]. In this study, ten local body parts were selected according to Houdas [43]. As illustrated in Figure 2a, the local body parts selected were forehead, left chest, belly, left back, right upper arm, left forearm, right hand back, left thigh, left calf, and left foot back. The mean skin temperature of each participant was obtained using the following general formula:

$$T_{skin} = f_1 \cdot T_1 + f_2 \cdot T_2 + \cdots + f_n \cdot T_n \tag{1}$$

where $T_{skin}$ is the mean skin temperature, °C; $T_i$ is the local skin temperature of each body part, °C; $f_i$ is the corresponding weighting factor.

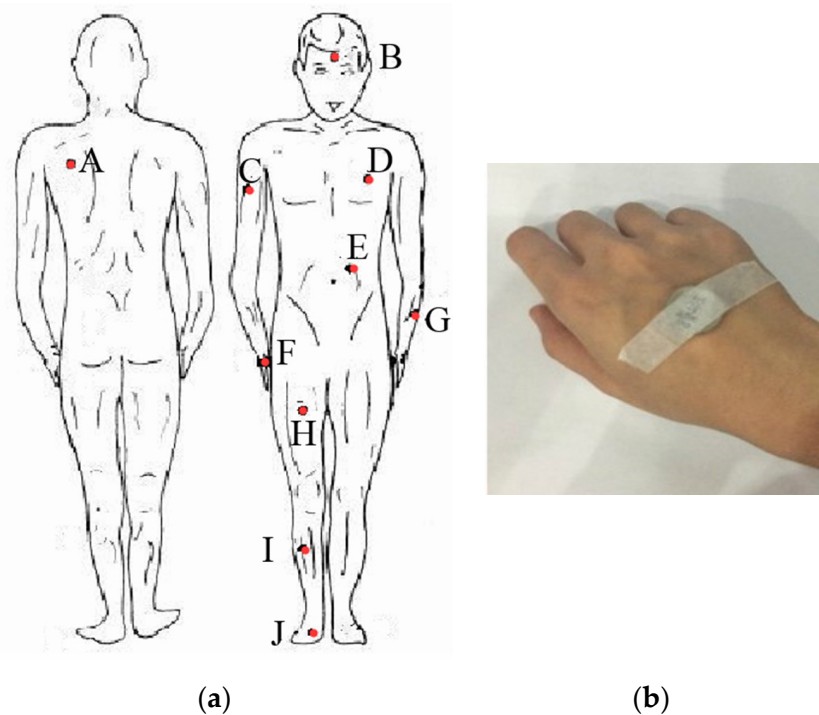

(**a**)          (**b**)

**Figure 2.** Skin temperature measurement. (**a**) Measured local body parts. (A: left back, B: forehead, C: right upper arm, D: left chest, E: belly, F: right hand back, G: left forearm, H: right thigh, I: right calf, J: right foot back); (**b**) IButton placement.

The corresponding weighting factors for the ten body parts were cited from Houdas [43] and they are listed in Table 3. The local skin temperature was recorded by attaching the IButton (DS1922L, Wdsen, Shanghai, China) to the skin surface of the participants, as illustrated in Figure 2b. Specific parameters of the IButton are listed in Table 2.

**Table 3.** The weighting factors of the ten body parts for mean skin temperature calculation.

| Forehead | Chest | Belly | Back | Upper Arm | Forearm | Hand Back | Thigh | Calf | Foot Back |
|----------|-------|-------|------|-----------|---------|-----------|-------|------|-----------|
| 0.06 | 0.12 | 0.12 | 0.12 | 0.08 | 0.06 | 0.05 | 0.19 | 0.13 | 0.07 |

2.3.3. Subjective Measurements

The participants' subjective assessment to the ambient environment was carried out by questionnaires that included questions on thermal sensations and their intentions of turning off the air conditioning. The participants' thermal sensation was assessed using the ASHRAE 7-point scale [44] shown in Figure 3 ($-3$ = cold, $-2$ = cool, $-1$ = slightly cool, 0 = neutral, +1 = slightly warm, +2 = warm, +3 = hot). The participants' intentions of turning off the air conditioning were obtained by answering the question: do you want to turn off the air conditioning now?

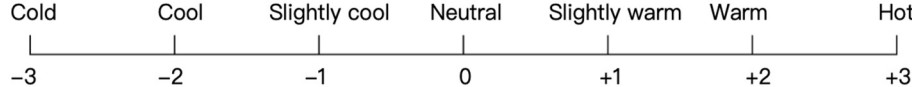

**Figure 3.** Thermal sensation 7-point scale of ASHARE.

*2.4. Experimental Procedures*

The detailed experimental procedure and environments are shown in Figures 4 and 5. As discussed above, four scenarios (A: 29→27 °C, B: 29→25 °C, C: 29→23 °C and D: 27→23 °C) were selected. Taking scenario A (29→27 °C) as an example, participants arrived

at the experimental room after the indoor temperature was set to 29 °C, then IButtons were attached to the skin of the ten body parts of the participants to monitor the skin temperatures. After that, all the participants were informed of the experimental procedure. This preparation time was approximately 20 min. After preparation, participants stayed in 29 °C for 30 min, during which questionnaires regarding overall thermal sensation were completed according to the 7-point ASHRAE sensation scale every 10 min. Then, the indoor temperature was reduced gradually to 27 °C within 30 min, during which questionnaires about overall thermal sensation were completed at intervals of 5 min, and the participants' intention of turning off air conditioning was recorded every 10 min. Finally, the room temperature was stabilized for the last 60 min and the questionnaires were filled in at intervals of 5 min. During the experiment, participants sat in a chair reading, surfing the internet, etc. Two groups of tests were performed each day, and each participant was allowed to participate only in one experiment per day to avoid the effect of frequent experiments on the experimental results. All 29 individuals completed the experiments of the four scenarios described above.

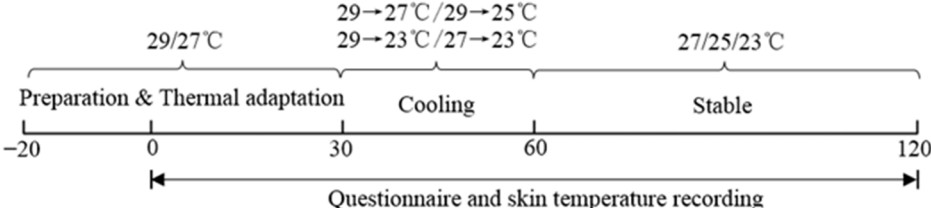

**Figure 4.** Experimental procedure.

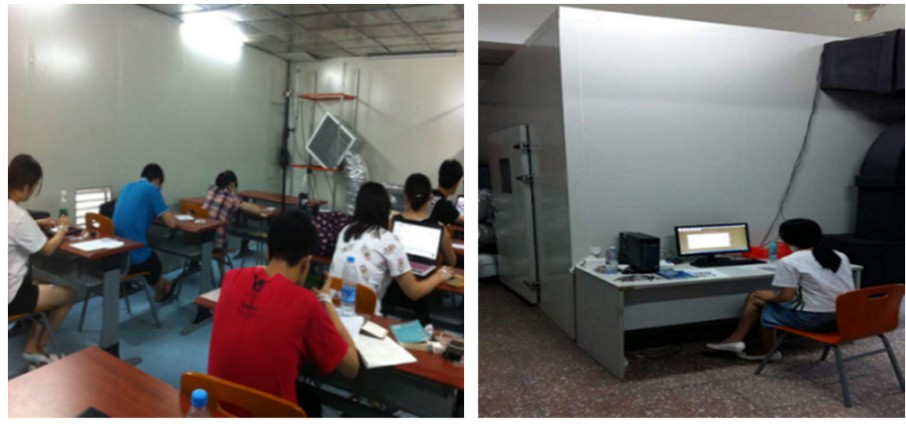

**Figure 5.** Internal and external view of the experimental chamber.

*2.5. Statistical Analysis*

Several statistical methods were performed in this study to analyze the collected data. Whether the data of the local skin temperatures followed normal distribution was checked using the Schapiro–Wilk test. Due to the normal distribution of skin temperature, the difference of skin temperature under different TSV was investigated using two-sample *t*-tests and two paired sample *t*-tests, and a difference of $p < 0.05$ was considered to be significant. The difference in mean skin temperature at 30 min and when air conditioning behavior intention occurred were analyzed by using two paired sample *t*-tests. A power exponent regression was adopted to describe the relationship between air conditioning duration and the probability of the participants to turn off the air conditioning.

## 3. Results

*3.1. Mean Skin Temperature and TSV*

Dynamic changes were observed in the mean skin temperature and thermal sensation with the variation in ambient temperature. As presented in Figures 6 and 7, after the

ambient temperature was stabilized, the mean skin temperature and thermal sensation decreased over a period of time.

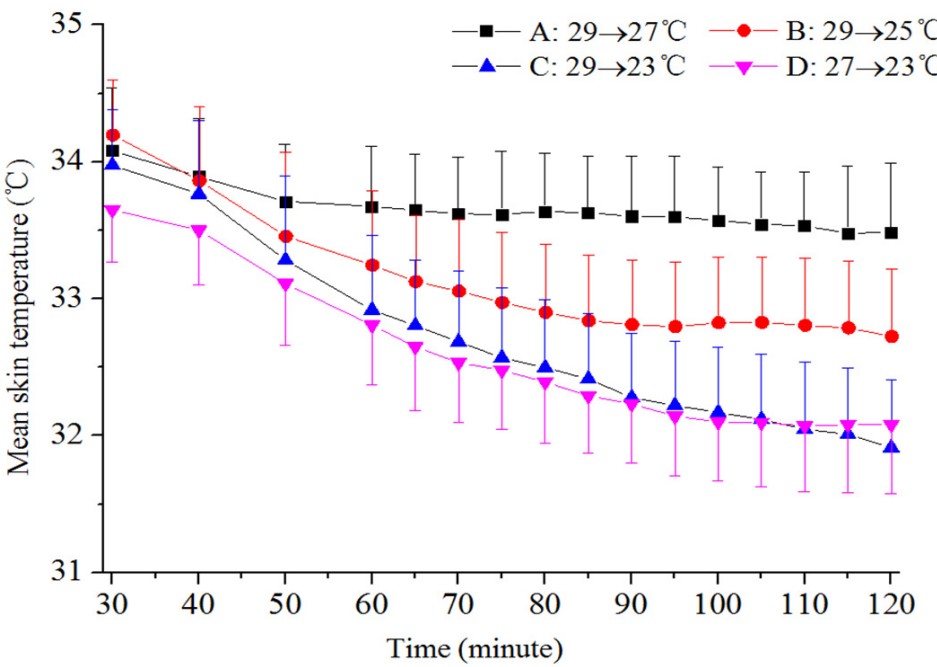

**Figure 6.** Mean skin temperature during scenarios A, B, C and D (mean ± SD).

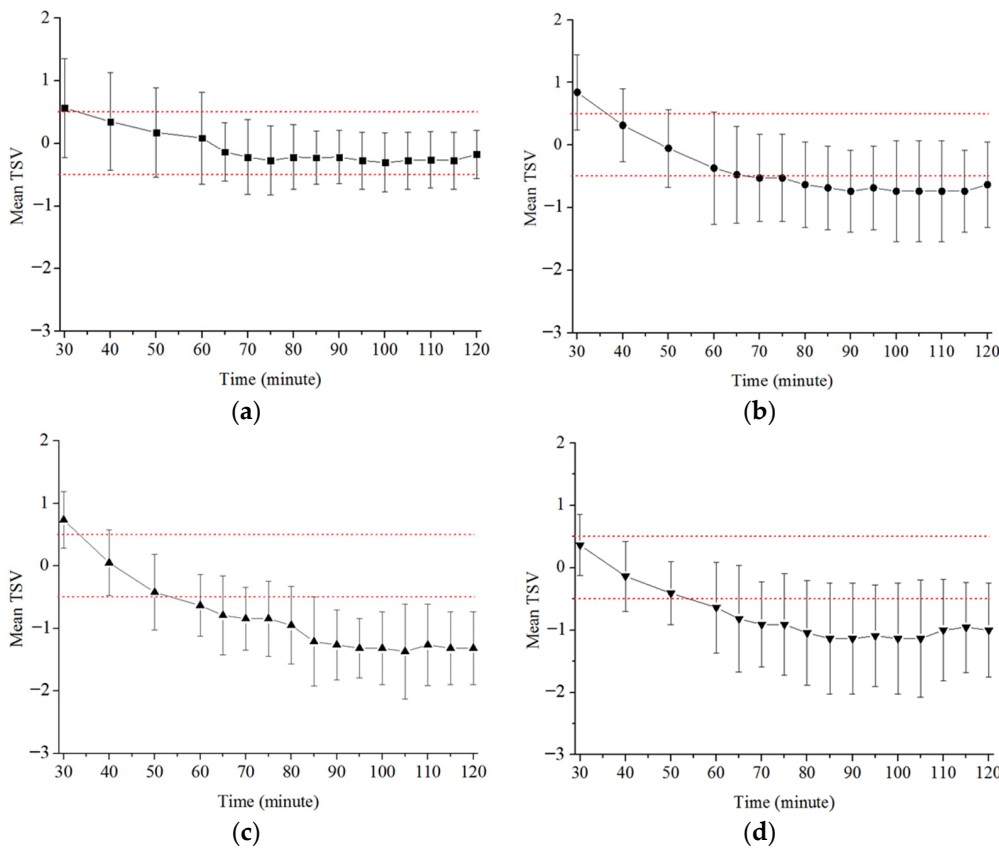

**Figure 7.** Mean thermal sensation vote during four experimental scenarios (mean ± SD). (**a**) Scenario A: 29→27 °C. (**b**) Scenario B: 29→25 °C. (**c**) Scenario C: 29→23 °C. (**d**) Scenario D: 27→23 °C.

The difference in skin temperature after stabilization between scenario A (29→27 °C) and B (29→25 °C) was approximately 0.7 °C, and it was almost the same between scenario B (29→25 °C) and C (29→23 °C). For scenario C (29→23 °C) and D (27→23 °C), the difference in mean skin temperature was significant at the initial ambient condition, but it quickly decreased and was almost negligible after the room temperature was stabilized.

The average TSV results for scenario A (29→27 °C) (Figure 7a), were within the range of ±0.5, which was considered as a thermally acceptable environment [44]. However, as shown in Figure 7b–d, for scenarios B (29→25 °C), C (29→23 °C), and D (27→23 °C), respectively, after 60 min of the experiments, the average TSV results were mostly out of the thermally acceptable range of the human body. Furthermore, the variation in thermal sensation represented by the vertical bar among participants in scenario A was much smaller than that in scenarios B–D, indicating smaller individual differences of thermal sensation under scenario A. Only two out of twenty-nine participants reported their intention of turning off the air conditioning during the experiment for scenario A.

### 3.2. Comparison of TSV When Turning off Air Conditioning

We found that the participants' thermal sensation could be different when they reported their intention of turning off the air conditioning. Some of the participants tended to turn off the air conditioning when TSV = −1 (slightly cool), while some other participants preferred turning off the air conditioning when TSV = −2 (cool). As an illustration example, Figure 8 shows the variations in the thermal sensations and marks the time when the participant's intention of turning off air conditioning occurred for participants A and B. Thermal sensations of the two participants both decreased at the beginning and then stayed constant. The difference was that the TSV = −1 (slightly cool) for participant A and the TSV = −2 (cool) for participant B when they reported their intentions of turning off the air conditioning.

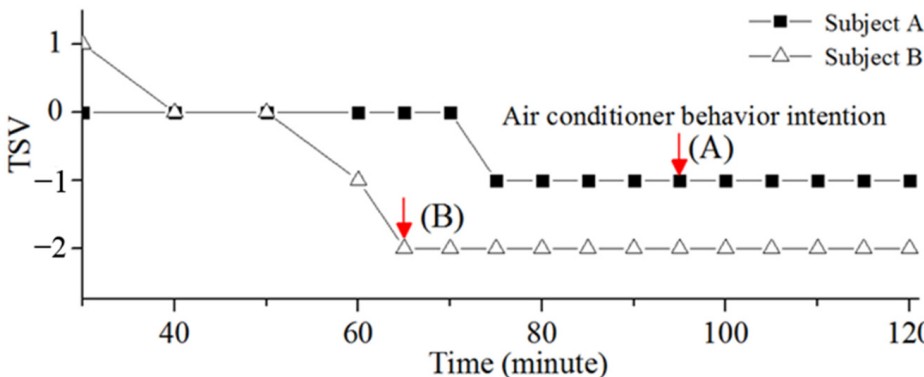

**Figure 8.** TSV and air conditioning off behavior intention of two typical participants in scenario D (27→23 °C).

Accordingly, we divided the participants into two categories (Class I and Class II) based on the participants' thermal sensations reported at the moment when their intention of turning off air conditioning arose. Class I included those who intended to turn off air conditioning when TSV = −1, and Class II included those corresponding to TSV = −2. The number of participants of Class I and Class II in four scenarios were shown in Figure 9. It also should be noticed that the participants that were analyzed were derived from the participants who wanted to turn off the air conditioning after the 30 min. The participants with the intention of turning off the air conditioning at/before the 30 min were excluded in this study.

It can be seen that the number of participants who were observed with the intention of turning off the air conditioning increased with the decrease in the ambient temperature, with 2, 11, 17, and 19 participants in scenarios A, B, C, and D, respectively. This can be illustrated by the fact that tolerance of the human body to cool temperature environments

decreases as the ambient temperature decreases. The number of participants of Class I was less in scenario C (29→23 °C) than in scenario D (27→23 °C). The reason can be explained as the initial temperature of 27 °C enhanced the cold feeling of humans under scenario D.

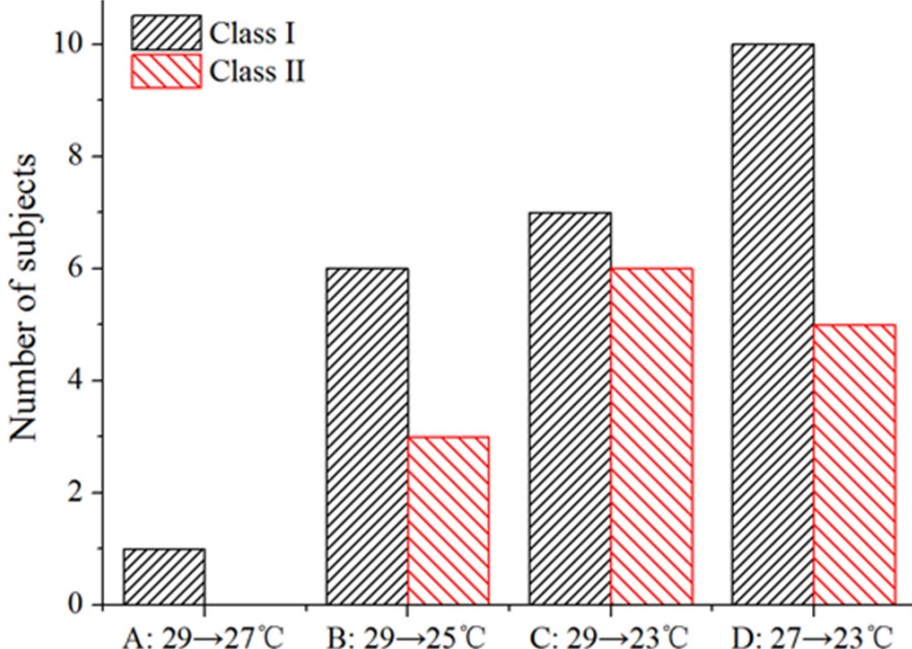

**Figure 9.** Number of participants who have the intention to turn off the air conditioning and their TSV when the air conditioning off behavior intention occurred after T = 30 in four scenarios (Class I: intent of turning off air conditioning occurred when TSV = −1, Class II: intent of turning off air conditioning occurred when TSV = −2).

Figures 8 and 9 illustrate that the thermal sensation when thermal dissatisfaction with the air conditioning environment occurred is different between Class I and Class II. In the four scenarios, the number of participants in Class I was more than that in Class II. This indicates that more participants showed a willingness to turn off the air conditioning when they felt slightly cool.

*3.3. Comparison of Skin Temperature When Turning off Air Conditioning*

Classifying participants according to their thermal sensations reported at the moment when their intention of turning off the air conditioning arose, helps not only make better predictions of thermal sensation [19], but also understand the occupants' adaptive behaviors. In this section, the difference in the mean skin temperature of participants who reported their intention of turning off the air conditioning between 30 min (moment 1) and when behavior intention of turning off air conditioning occurred (moment 2) were analyzed by using two paired sample *t*-tests. Scenario A was excluded due to the small sample size. Figure 10 illustrates that the mean skin temperature of participants between moment 1 and 2 was significantly different. The occupants' behavior of turning off air conditioning related to their thermal sensation and skin temperature in scenario C and scenario D were analyzed by using two-sample *t*-tests. Scenario A and scenario B were excluded due to the small sample size. Figure 11 illustrates the relationship between the mean skin temperature and thermal sensation for participants of Class I and Class II in experimental scenarios C and D, respectively. No significant difference was found in the mean skin temperature when the participants expressed willingness to turn off the air conditioning between Class I and Class II.

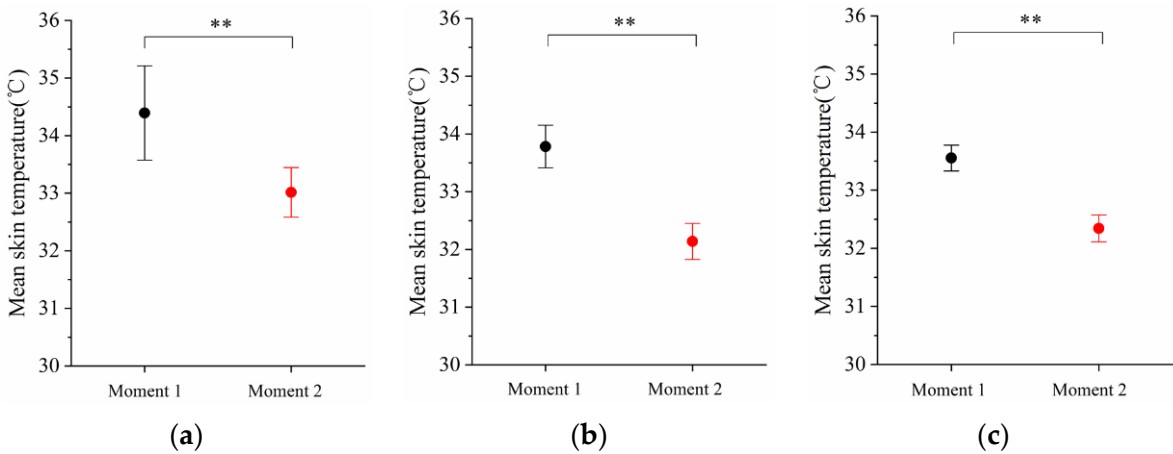

**Figure 10.** Interval plot of mean skin temperature by moment 1 (at 30 min) and moment 2 (when behavior intention of turning off air conditioning occurred) in scenario B, C, and D. ((**a**): scenario B, 29→25 °C, (**b**): scenario C, 29→23 °C, (**c**): scenario D, 27→23 °C). Vertical bars represent 95% confidence intervals (CIs) for the mean, and ** denotes a significant difference between two moments ($p < 0.05$).

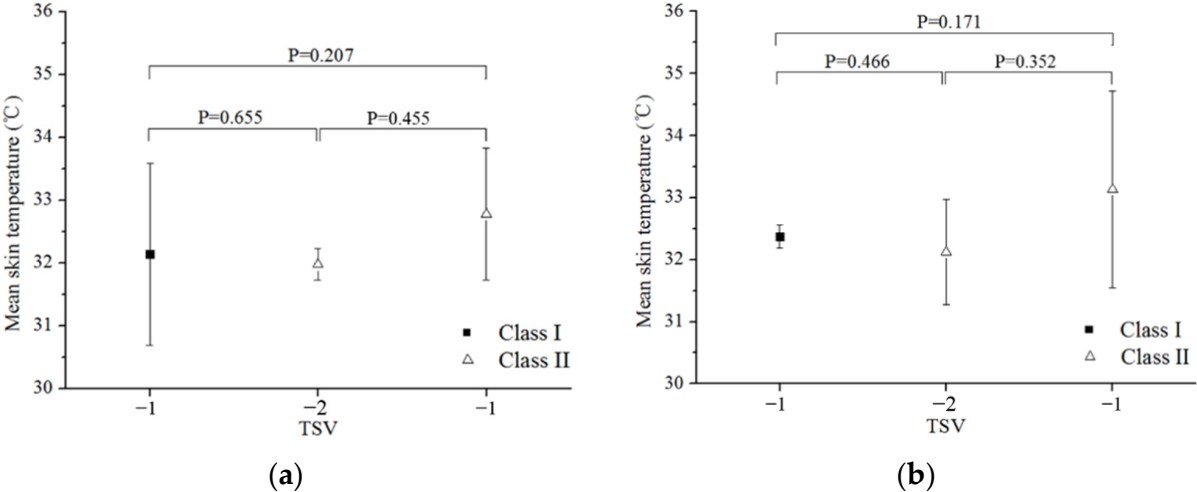

**Figure 11.** Interval plot of mean skin temperature by thermal sensation of Class I and Class II in scenario C and D. ((**a**): scenario C, 29→23 °C, (**b**): scenario D, 27→23 °C). Vertical bars represent 95% confidence intervals (CI) for the mean.

Further, local skin temperatures of ten body parts for scenario C are shown in Figure 12. No significant difference was found in the skin temperature of forehead, chest, belly, back, upper arm, and thigh between Class I (TSV = −1) and Class II (TSV = −2). However, for the hand back, forearm, calf, and foot, no significant difference was found in skin temperature when the participants wanted to turn off the air conditioning (TSV = −1 for Class I and TSV = −2 for Class II), with $p = 0.693$, $p = 0.124$, $p = 0.721$, and $p = 0.700$ in hand back, forearm, calf, and foot back, respectively; whereas, a significant difference was found in the skin temperature of these four body parts when participants of Class I and Class II both felt slightly cool (TSV = −1). Similar results were obtained in scenario D (Table 4), in which a significant difference was found in skin temperature of foot and calf between Class I and Class II when TSV = −1, with $p = 0.008$ and $p = 0.012$ in foot and calf, respectively. However, no significant difference of skin temperature was observed with $p = 0.911$ and $p = 0.219$ in foot and calf when participants of Class I felt slightly cool (TSV = −1) and participants of class II felt cool (TSV = −2).

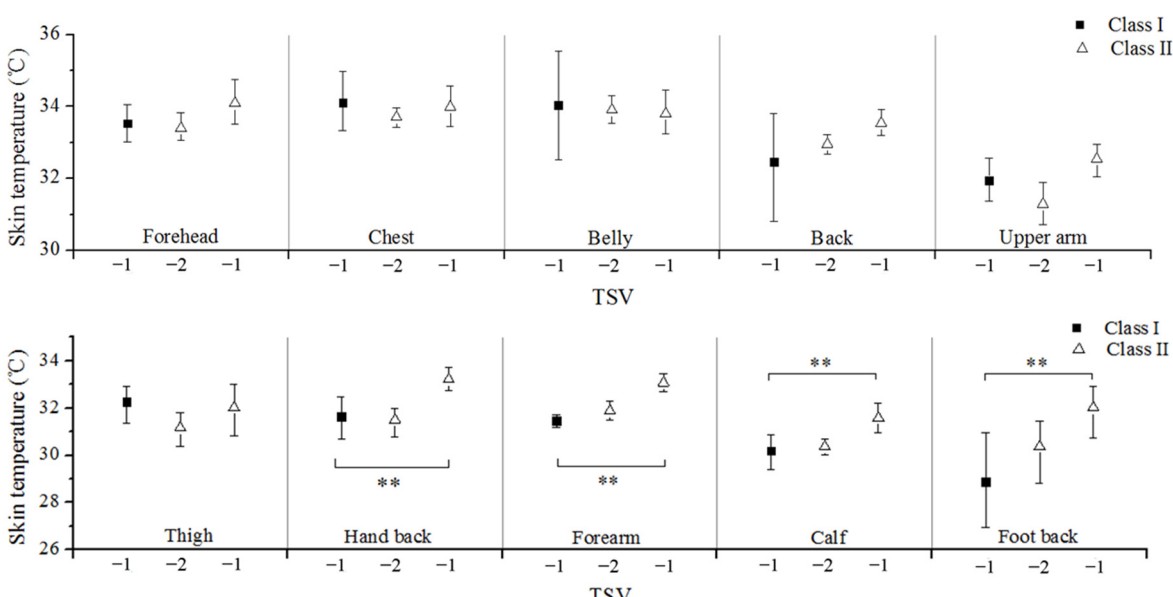

**Figure 12.** Interval plot of local skin temperature by thermal sensation of ten body parts of Class I and Class II in scenario C (29→23 °C). Vertical bars represent 95% CI for the mean, and ** denotes a significant difference between two TSV values (*p* < 0.05).

**Table 4.** *p*-value of two-sample *t*-test of local skin temperature of ten body parts of Class I and Class II for scenario C and D.

| | Scenario C: 29→23 °C (TSV of Class I vs. Class II) | | Scenario D: 27→23 °C (TSV of Class I vs. Class II) | |
|---|---|---|---|---|
| | −1 vs. −1 | −1 vs. −2 | −1 vs. −1 | −1 vs. −2 |
| Forehead | 0.239 | 0.333 | 0.451 | 0.713 |
| Chest | 0.886 | 0.329 | 0.289 | 0.979 |
| Belly | 0.834 | 0.938 | 0.741 | 0.939 |
| Back | 0.238 | 0.913 | 0.546 | 0.871 |
| Upper arm | 0.176 | 0.349 | 0.458 | 1.000 |
| Thigh | 0.723 | 0.406 | 0.330 | 0.776 |
| Hand back | 0.034 ** | 0.693 | 0.216 | 0.253 |
| Forearm | 0.000 ** | 0.124 | 0.205 | 0.454 |
| Calf | 0.034 ** | 0.721 | 0.012 ** | 0.911 |
| Foot back | 0.047 ** | 0.700 | 0.008 ** | 0.219 |

** indicates statistical significance (*p* < 0.05).

### 3.4. Logistic Regression of Duration vs. the Probability of Turning off Air Conditioning

Through the statistics of the probability of participants who are willing to turn off the air conditioning under four experimental scenarios, the probability of turning off air conditioning at a series of specific time points was obtained. Based on that, a power exponent regression model, similar to the form of the logistic regression equation was adopted to describe the relationship between air conditioning duration and the probability for the participants to turn off the air conditioning. The fitting equation used is as follows:

$$p(t) = \frac{1}{1 + a^{-(t-\theta)}} \tag{2}$$

where *p* is the probability of turning off air conditioning; *t* is air conditioning duration; and *a* and *θ* are the characteristic parameters.

Figure 13 illustrates the fitting curves between the percentage of people who want to turn off the air conditioning and air conditioning duration in three scenarios B (29→25 °C),

C (29→23 °C), and D (27→23 °C) based on the logistic regression, and the characteristic parameters are summarized in Table 5. The results of scenario A (29→27 °C) were omitted due to the small sample size, only two out of twenty-nine participants reported their intentions of turning off the air conditioning. This indicates that the participants would probably not turn off the air conditioning in an air conditioning environment with an ambient temperature more than 27 °C. However, in the case of an ambient temperature below 27 °C, as the air conditioning duration increased, the number of participants who wanted to turn off the air conditioning increased. The fitting curves of scenarios C (29→23 °C) and D (27→23 °C) were similar (a = 1.04, θ = 60.37 of scenario C and a = 1.04, θ = 61.76 of scenario D), whereas a significant difference was shown between scenario B (29→25 °C) and scenario C (29→23 °C). Participants had a much higher probability of turning off the air conditioning in scenario C than in scenario B after the air conditioning was on although the two scenarios had the same initial indoor temperature of 29 °C. This indicates that the temperature set point strongly influenced when participants tended to turn off the air conditioning, and therefore determined the air conditioning duration. Based on the logistic regression model (Figure 13), approximately 85% of the participants were expected to turn off the air conditioning after the air conditioning was continuously on for approximately 100 min at the ambient temperature set point 23 °C, or after the air conditioning was continuously running for approximately 150 min at the ambient temperature set point 25 °C.

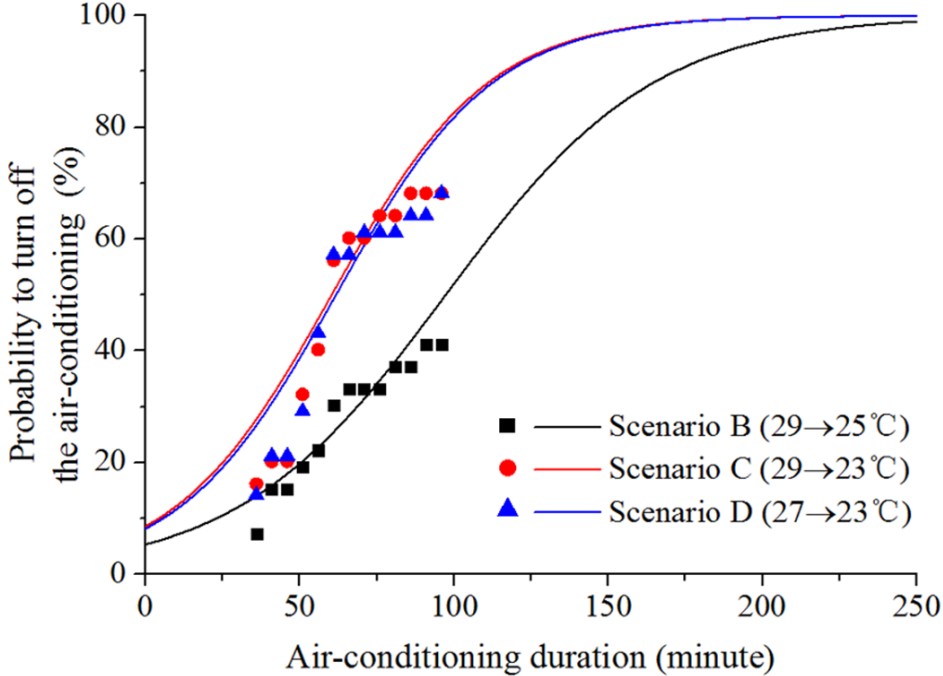

**Figure 13.** The probability to turn off the air conditioning as a function of the air conditioning duration. (The scatter points represent the probability with air conditioning off intention in scenarios B–D).

**Table 5.** Results of the characteristic parameters of logistic regression for three experimental scenarios.

| Experimental Scenario | *a* | *θ* | R$^2$ | *p* |
|---|---|---|---|---|
| B: 29→25 °C | 1.03 | 97.34 | 0.888 | <0.001 |
| C: 29→23 °C | 1.04 | 60.37 | 0.883 | <0.001 |
| D: 27→23 °C | 1.04 | 61.76 | 0.855 | <0.001 |

## 4. Discussion

### 4.1. Physiological Driving Force for Turning off Air Conditioning

We found that the TSV changed when the participants' intentions of turning off the air conditioning arose (TSV = −1 for Class I, and TSV = −2 for Class II); however, there was no significant difference in the local skin temperatures of exposed body parts between Class I and Class II, such as foot and calf. Therefore, the occupants' behavior of turning off air conditioning is largely dependent on the local skin temperature of exposed body parts, especially foot and calf. This observation aligns with findings from our previous experiments [45], which were conducted in an actual air conditioning room. The skin temperature of six body parts (sole, neck, chest, hand, forearm, and thigh) and thermal sensation for 11 participants in an actual air conditioning environment were investigated. The results from this study showed that the sole temperature was a vital parameter to interpret the behavior of turning off air conditioning (thin pants, thin short-sleeved shirts, and slippers, approximately 0.39 clo).

In addition, as shown in Figure 10, the results of the two paired sample *t*-test showed that the mean skin temperature of participants who reported their intention of turning off the air conditioning was significantly different when at 30 min and when the behavior intention of turning off the air conditioning occurred, with $p = 0.000$ in scenario B, C, and D (scenario A was excluded due to the small sample size). From the physiological perspective, human perception of environmental hot and cold depends on skin temperature, which is also an indispensable parameter for predicting thermal sensation [46,47]. However, in these studies the effect of skin temperature and thermal sensation on regulatory behavior was not addressed. This study established an association between skin temperature, thermal sensation, and air conditioning behavior, and clarified the direct effect of skin temperature on the behavior of turning off air conditioning. The results of this study were similar to the experimental results in Song's study [35] that was conducted in a classroom, and Vargas's study [36] of which the environment temperature was set at $32 \pm 1\,°C$ and $42 \pm 1\,°C$, and supports the viewpoints that skin temperature plays an important role in the occupants' behavior. On the other hand, as shown in Figure 11, no significant difference was found in the mean skin temperature for Class II between slightly cool (TSV = −1) and cool (TSV = −2). This may due to the following reasons: (1) the time required for TSV changes was not sufficient to cause a significant difference in skin temperature; and (2) though rapid drops in skin temperature can be seen in exposed body parts, the weight of the skin temperature of these body parts in the mean skin temperature calculation formula is relatively small, totally 0.31, which weakened its contribution to the mean skin temperature, while the skin temperature of core body parts, which was relatively stable, dominated the calculation of the mean skin temperature.

Thermal adaptation in the built environment can be attributed to three different processes: behavioral adjustment, physiological acclimatization, and psychological habituation or expectation [48]. However, in the climate chamber experiment in this study, the participants had no more adaptation behavior other than turning off the air conditioning, nor did they have concerns about the operation cost of the air conditioner. The behavior of turning off the air conditioning was completely determined by their physiological adaptation state.

### 4.2. Environmental Driving Force for Turning off Air Conditioning

From the environmental perspective, in the current study on air conditioning behavior, the establishment of an air conditioning use probability model based on indoor [11,28,29] or outdoor [29,30,32] temperatures has been the focus of research. As shown in Figure 14, the research of Ren [28] and Tanimoto [29], based on the continuous monitoring of actual dwellings, indicated that the probability of air conditioning behavior was exponentially increased along with the increase in the indoor temperature (Figure 14a) or outdoor air temperature (Figure 14b), highlighting the important role of environment on air conditioning behavior.

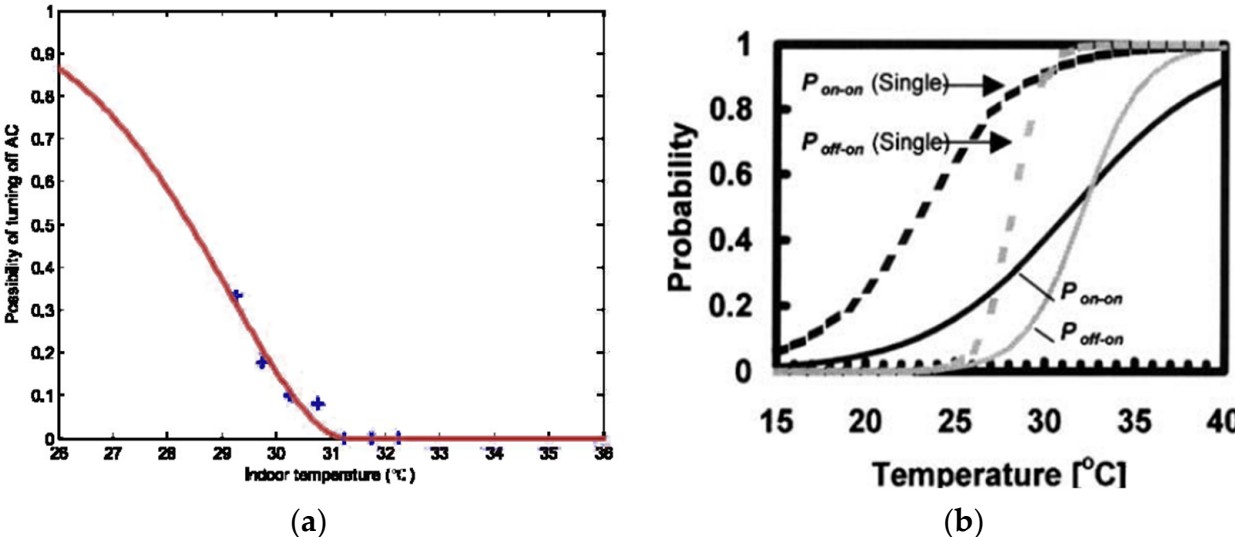

**Figure 14.** Air conditioning behavior models in other researches. (**a**) Ren's research [28]; (**b**) Tanimoto's research [29].

Interestingly, according to the fitted curve in Figure 13, we found that when the ambient set temperature is provided, the probability of turning off the air conditioning exponentially increased along with the increase in air conditioning duration. As an illustration, approximately 50% of the participants were expected to turn off the air conditioning after the air conditioning was continuously on for 100 min at the ambient temperature 25 °C, whereas the probability of turning off the air conditioning would reach 85% after turning on air conditioning for approximately 150 min also at the ambient temperature 25 °C. That is, the air conditioning duration also affects the probability of the behavior to turn off the air conditioning. Similar findings have been confirmed from some other research of thermal comfort. Tham's study [49] showed that longer exposure to the indoor temperature of 20.0 °C led to cooling sensations due to lower skin temperatures ($p < 0.0001$) and was perceived as the least comfortable. Additionally, regression curves of the behavior of turning off the air conditioning can be different when air conditioning set temperature is different. Regression results in Figure 12 show that it would take approximately 100 min for 85% of the participants to turn off air conditioning at the ambient temperature 23 °C, approximately 50 min shorter in comparison with the situation at the ambient temperature 25 °C. A corresponding fit could not be made for the ambient temperature of 27 °C, indicating that it would take a longer time for 85% of the participants to turn off the air conditioning. Therefore, air conditioning duration, together with the ambient temperature, affect the probability of the behavior of turning off the air conditioning. In this regard, we can confirm that the lower the ambient temperature or the longer the air conditioning environment is maintained, the stronger the willingness of the participants to turn off the air conditioning. More specifically, 27 °C appeared to be an appropriate air conditioning set temperature for the typical clothing style (thin T-shirt, thin shorts, and slippers) in Chinese residential buildings in summer. In this environment, the occupants may not feel cold discomfort and develop a desire to turn off the air conditioning.

It is worth noting that the discussion in this section focuses on the situation that air conditioning behavior occurred when participants felt cool. Event triggering air conditioning behavior, such as turn off when leaving, is not within the scope of the discussion. In this way, this study highlighted the impacts of environments from two dimensions, that is both environment stimuli itself and environment exposure time, affecting the air conditioning duration.

*4.3. Limitations*

There are still some limitations in this study. First, participants were required to report their intentions of turning off the air conditioning instead the actual behaviors of turning off the air conditioning during the experiment. Second, many studies have proposed the feasibility of local cooling and heating devices to improve human thermal comfort, for example heated seats [50], and local cooling of head, chest, and back [51]. This paper only focused on the relationship among skin temperature, thermal sensation, and air conditioning behavior under a uniform ambient environment, more experiments should be carried out where participants are provided with local cooling and heating devices in a future study. Third, previous studies have shown that individual differences may impact the thermal sensation prediction of the human body [21,52–56], but the effect of individual differences in turning off the air conditioning remains unknown, factors such as sex, BMI, and age should be taken into consideration in the process of selecting survey participants. Fourth, this study was only carried out in Beijing, China, and can be extended to warmer regions in China and to a larger sample size.

**5. Conclusions**

In this study, a series of experiments were conducted in an environmental chamber aiming to understand the behavior of turning off air conditioning by associating it with environmental conditions, the human physiological state, and thermal sensations. The findings are summarized as follows:

(1) Skin temperature and thermal sensation changed dynamically in an air conditioning environment. After the ambient temperature became stable, skin temperature and thermal sensation might still change over a period of time. The time required for the skin temperature and the thermal sensation to remain unchanged increased as the ambient temperature set point decreased;

(2) The participants' thermal sensations when they expressed their thermal dissatisfaction with the air conditioning environment and reported their intentions of turning off the air conditioning was different, by which the participants were divided into two categories: Class I and Class II. Participants in Class I tended to turn off the air conditioning when they felt slightly cool (TSV = −1), while the participants in Class II tended to turn off the air conditioning when they felt cool (TSV = −2). However, there was no statistically significant difference in skin temperature of exposed body parts, especially in foot and calf, between the participants of Class I and Class II when they reported their intentions of turning off the air conditioning. The behavior of turning off the air conditioning is thus largely dependent on the local skin temperature of exposed body parts;

(3) The probability of turning off air conditioning exponentially increased along with the increase in the air conditioning duration. Additionally, regression curves of the participants' behavior of turning off air conditioning was different when air conditioning set temperature was different. Therefore, air conditioning duration, together with the ambient temperature, affect the probability of the behavior of turning off air conditioning. The lower the ambient temperature or the longer the exposure to the air conditioning environment, the stronger the willingness of the participants to turn off the air conditioning.

This study enriches the understanding of the behavior of turning off air conditioning from both physiological and environmental perspectives, and contributes to avoiding overestimating or underestimating building energy consumption. Furthermore, this study also provides a preliminary theoretical basis for the establishment of a control strategy that determines air conditioning switches according to the physiological state of the occupants, and such a control strategy would be very meaningful for institutions such as kindergartens and nursing homes.

**Author Contributions:** Conceptualization, Y.J.; methodology, Y.J. and S.L. (Shuwei Liu); formal analysis, Y.J., S.L. (Shuwei Liu) and S.L. (Shengjie Liu); investigation, S.L. (Shuwei Liu) and S.L. (Shengjie Liu); resources, M.B. and Z.L.; data curation, S.L. (Shuwei Liu) and S.L. (Shengjie Liu); writing—original draft preparation S.L. (Shuwei Liu) and S.L. (Shengjie Liu); writing—review and editing, Y.J. and S.L. (Shuwei Liu); supervision, Y.J., M.B. and Z.L.; project administration, M.B. and Z.L. All authors have read and agreed to the published version of the manuscript.

**Funding:** This research was financially supported by the Opening Funds of State Key Laboratory of Building Safety and Built Environment and National Engineering Research Center of Building Technology (Grant No. BSBE2021-06) and the National Key Research and Development Program of China (Grant No. 2016YFC0700500).

**Institutional Review Board Statement:** This study was carried out in accordance with the recommendations of the Ethical Code of the Beijing University of Technology and of the Code of Ethics approved by the General Assembly of the Chinese Association of Psychology. All participants gave written informed consent in accordance with the Declaration of Helsinki. Ethical approval was not required for this study in accordance with the national and institutional guidelines.

**Informed Consent Statement:** Informed consent was obtained from all participants involved in the study.

**Data Availability Statement:** The data presented in this study are available on request from the corresponding author.

**Acknowledgments:** The authors would like to thank the participants who were involved in the investigation for their great cooperation.

**Conflicts of Interest:** We declare that we have no financial and personal relationships with other people or organizations that can inappropriately influence our work, there is no professional or other personal interest of any nature or kind in any product, service, and/or company that can be construed as influencing the position presented in, or the review of, the manuscript entitled "Climate chamber experiment study on the association of turning off air conditioning with human thermal sensation and skin temperature".

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
