# Peer review of "Climate Chamber Experiment Study on the Association of Turning off Air Conditioning with Human Thermal Sensation and Skin Temperature"

_buildings, doi:10.3390/buildings12040472_

Round 1

Reviewer 1 Report

I have reviewed this article which addresses some findings on skin temperature and behavioural actions about air conditioning.

The authors discuss an interesting problem whose solution could be adequate for improving the performance of air conditioning units and energy use.

To begin with, TSV Thermal Sensation Vote, is not properly defined.

The paper describes and employs the existing literature in an adroit manner. The methods of and preparation of the experiments are well defined and referenced.

A study hinting to the costs of energy use in China and income level of the participants would perhaps be useful.

The results’ part is well researched and shows an appropriate output.

The conclusions part of the study could be enhanced with the intentions to continue the research and the impact expected in HVAC systems. Also if the studies could be extended to warmer regions within China other than Beijing.

Summary of evaluation: This article is interesting and exposes a relevant question. I recommend publication after minor reviews in the sense exposed above.

Author Response

Point 1: To begin with, TSV Thermal Sensation Vote, is not properly defined.

Response 1: Thanks for your comments.

In the revised manuscript(Section 2.3.3. Subjective measurements), the TSV is defined as follows:

“The participants' thermal sensation was assessed using the ASHRAE 7-point scale [44] shown in Figure 3(-3=cold, -2=cool, -1=slightly cool, 0=neutral, +1=slightly warm, +2=warm, +3=hot).

Figure 3. 7-point thermal sensation scale of ASHARE.

Point 2: A study hinting to the costs of energy use in China and income level of the participants would perhaps be useful.

Response 2: Thanks for your comments.

Indeed, the cost of energy use and income level may be important potential factors influencing people's air conditioning behavior in real life. However, this experiment was conducted in a climate chamber and was designed to analyze the relationship between occupants’ behavior of turning off air conditioning and physiological state (e.g. mean skin temperature, local skin temperature) as well as thermal sensation, so that subjects were not disturbed by the economic factors mentioned above. Actually, The suggestions made by the reviewer are well worth conducting practical research in taking the findings of this paper to further generalization.

Point 3: The conclusions part of the study could be enhanced with the intentions to continue the research and the impact expected in HVAC systems.

Response 3: Thanks for your comments.

In the revised manuscript, we added the following description of the expected impact of this study on the HAVC system:

“This study also provides a preliminary theoretical basis for the establishment of a control strategy that determines air conditioning switches according to the physiological state of the occupants, and such a control strategy would be very meaningful for institutions such as kindergartens and nursing homes.”

Point 4: Also if the studies could be extended to warmer regions within China other than Beijing.

Response 4: Thanks for your comments.

In the revised manuscript, we have added to the Limitions section with the following details:

“Fourth, this study was only carried out in Beijing, China, and could be extended to warmer regions within China and to a larger sample size.”

Reviewer 2 Report

Overall this is an interesting study, however, more directed analyses could be completed to obtain more information about the behaviour of the participants.

One major limitation is only requiring participants to report every 10 minutes whether they wished the air conditioning to be turned off. Consequently, your findings may not be sensitive enough to make decisions regarding the use of air conditioning. 

Introduction

No comment.

Methods:

Be consistent between mins and minutes.

You explain participants were reading and surfing the internet during the scenarios – please explain whether the reading/surfing was their own choice or controlled in any way. Rationale for this request is due to the possible influence of the material read being a distraction from a focus on thermal perception.

Why was the question ‘Do you want to turn off the air conditioning now?’ asked every 10 minutes and not every 5 minutes like the thermal sensation or collected at the point the participant would like to express/action this behaviour. Please explain and be more specific about the limitations of this approach i.e. sensitivity of your experiment to identify this behaviour.

Did you randomise the order of the trials for participants?

Statistical analysis:

Your description of the statistical analyses used in this study needs to be more detailed in regard to what comparisons you are making and why; including an explanation of the use of the power exponent regression model to describe the relationship between air conditioning duration and the probability for the participants to turn off the air conditioning. In addition, please provide a rationale for using T-tests over multi-comparison tests such as ANOVA that would allow you make comparisons between scenarios and time points.

Results

In a quantitative paper generally only results of the study are described in the results section they are not discussed/explained. Currently, the results section includes elements of discussion such as below which would be better placed in the discussion.

‘Thus, 27℃ appeared to be a feasible air conditioning set temperature under the typical clothing style (thin T-shirt, thin shorts, and slippers) in Chinese residential building in summer. However, it is worth noting that the participants in this study are young people, and the thermal environmental demanded of children and the elderly may be slightly different’. This is a really good point but would be better placed in the discussion.

It would be useful to provide more details on the participants who were categorised as class I and class ii (i.e. sex, age, body surface area) if you have this information? These details may help explain any differences between the group and their perceptual response.

This following sentence does not make sense:

‘Furthermore, over 85.7% of participants in Class â…¡ were reported their intentions of turning off the air conditioning when their thermal sensation initially observed to be cool. This sentence does not make sense.’

Your results suggest that TSV is not a sensitive tool to detect the behaviour of wanting to turn off air conditioning and/or your interval of 10 minutes to record this information. However, changes in skin temperature appear to be more predictive. To reinforce this observation, it would be useful in Figure 7 to superimpose the mean skin temperature over the TSV votes.

Figure 9 also reinforces that skin temperature is the main driver for human behavioural regulation and questionnaires that assess human’s perception of their environment are less reliable as 1) humans interpret sensations differently and 2) these tools are sometimes not sensitive enough to pick up this behaviour.  

Why have you used Table 4 instead of figures similar to figure 10?

Please can you explain in more detail why participants with the intention of turning off the air conditioning at/before the 30 minutes were excluded in this study. Is there anything interesting about these participants i.e. did they want to turn off the air conditioning at similar skin temperatures? Did they record the same TSV?

Could you complete some sex and age comparisons?

Did you measure rate of skin temperature – if so is this measure any more predictive of reporting the desire to turn off the air conditioning?

The following sentence is a discussion point:  

‘Participants had a much higher probability of turning off the air conditioning in scenario C than that in scenario B after the air conditioning is on although the two scenarios had the same initial indoor temperature of 29℃. This indicates that past thermal experience difference had negligible impacts on the air conditioning duration. In contrast, the temperature set point, rather than initial indoor temperature strongly influenced when participants tended to turn off the air conditioning, and therefore determined the air conditioning duration.’

In addition, it might be useful to stress that it is more likely that participants will reach skin temperatures that are uncomfortable the longer the air conditioning is on especially at the lower set points?

Discussion:

Overall the discussion could be more to the point and more punchy in explaining the results and highlighting any comparisons with similar studies. It needs to be more concise and informative.

The discussion could be improved by incorporating information on previous work examining skin temperature and how it drives perception and behaviours. For example, how perceptions maybe influenced by rate of skin temperature change, set point of skin temperature (starting absolute skin temperature), final skin temperature, change in skin temperature – for example examine Hensel, H. (1981) Thermoreception and temperature regulation. Academic Press London and Zhang, H (2003) Human thermal sensation and comfort in transient and non-uniform thermal environments. PhD Thesis. University of California, Berkley, CA, 94720-1839, USA.

This sentence does not make sense-

‘In addition, for the mean skin temperature, on the one hand, results of the two paired sample t-test showed that the mean skin temperature of participants who were reported their intention of turning off air conditioning was significantly different when at the 30 minute and when behaviour intention of turning off air conditioning occurred,’

Figure 9 does not demonstrate this observation advised on page 12 – The figure indicates that comparisons were only made between class i and class ii, not between class ii.

‘On the other hand, as shown in Figure 9, no significant difference was found in mean skin temperature for Class II between slightly cool (TSV=-1) and cool (TSV=-2).’

The following paragraph is not clear in regard to the message you are trying to portray. The paragraph could be more concise.

From the environmental-driven perspective, in the current study on air conditioning behavior, the establishment of air conditioning use probability model based on indoor [11, 28, 29] or outdoor [29, 30,32] temperatures have been the focus of research. The researches of Ren’s [28] and Tanimoto’s [29] are shown in Figure 12 (a) and Figure 12 (b) in the form of regression curves. In the two researches, based on the continuous monitoring of actual dwellings, the probability of air conditioning behavior was exponentially increased along with the increase of the indoor temperature (Figure 12 (a)) or outdoor air temperature (Figure 12 (b)), highlighting the important role of environment on air conditioning behavior. For example, in Ren’s study, the fitted probability to turn off the air conditioning of one apartment bedroom was more than 85% when the indoor temperature was 25℃ and less than 40% at indoor temperature 29℃

Conclusions could be punchier. It would also be useful to include advise on how the findings could be used to inform policy regarding the use of air conditioning and reducing carbon costs. 

Be careful with this following statement:

When the air conditioning temperature set point was 27℃, mean TSV of the participants was within the range of ±0.5, besides, only 2 out of the 29 participants were reported their intention of turning off the air conditioning in this temperature environment. Thus, 27℃ could be regarded as an acceptable indoor temperature to occupants under the clothing style (thin T-shirt, thin shorts, and slippers) in Chinese residential building in summer.

What about a temperature set point of 29℃?

Reviewer 3 Report

Recommendations 

The paper is well written and presented. The scope of study is well-established to contribute on thermal comfort studies globally. Therefore, the paper is required additional work in order to improve the credibility of this rigorous research work. I outlined my recommendations as follow:

  1. Improve Abstract to give less context and more on the knowledge gap, research questions, methods and key findings
  2. In Introduction section, please outline the main aim, objectives and research questions clearly and articulate the research questions to implement the neutral adaptive thermal comfort thresholds as an output of the experiments conducted in the climate chamber
  3. Novelty of the study should be explained
  4. The authors have been discussed the previous scholars’ work in the Introduction but this is not sufficient to support the research outcomes presented in the Results section. I recommend to the authors to open-up a new section and consider these literature types as follows; systematic literature review or comprehensive literature review to study worldwide literature on thermal comfort studies 
  5. I recommend to the authors to the use this open-source software to conduct the systematic literature review on thermal comfort studies effectively. Here is the link of the open-source software tool - https://www.vosviewer.com - The authors generated the selected keywords and import the data into this software which allows the researcher to generate the visual maps. I believe that this tool could increase the scientific credibility of their research work
  6. Methodology section should be re-conceptualised, the authors should provide more detail on technical specifications of the climate chamber, image of the climate chamber and also image of the laboratory environment should be presented 
  7. Methodology section should refer more similar pilot studies to demonstrate the significance of authors’ their own work. I recommend to the authors to read this article - Ozarisoy, B., & Altan, H. (2021). Regression forecasting of ‘neutral’ adaptive thermal comfort: A field study investigation in the south-eastern Mediterranean climate of Cyprus. Building and Environment, 202. https://doi.org/10.1016/j.buildenv.2021.108013 - To increase the credibility of the authors’ their own work, I recommend the authors to cite this article while they are referring their own methodological framework in thermal comfort studies 
  8. The statistically representativeness of the sample size should be discussed in the Methodology. Is this sample size sufficient to make a generalisation of the study findings in thermal comfort studies? 
  9. The method should give an honest appraisal of how the sample size were chosen and reference the work of others who have developed statistically representative archetypes 
  10. 10. With regards to the identify the statistically representativeness of the sample size, please consider the statistical power of the survey. Use this open-source tool to identify the appropriate type of statistical method. Here is the link - https://www.psychologie.hhu.de/arbeitsgruppen/allgemeine-psychologie-und-arbeitspsychologie/gpower - This is the power analysis tool which helps you to identify the appropriate type of statistical method. Please use this tool and revise the statistical results presented in the Results section. 
  11. I recommend to the authors’ to conduct Fisher’s exact test to analyse the data related to the Likert scale findings presented in this paper. 
  12. To assess the thermal comfort parameters, the authors should conduct Pearson’s correlations for the accuracy of the findings. A good summary of appropriate techniques is given in Khamis (2008) Measures of Association – which to choose? Journal of Diagnostic Medical Sonography 24: 155-162. 
  13. Whilst correlations are indicative of association, there is scope with the data to do hypothesis testing of significant differences between variables (e.g. using contingency tables) and would add weight to the results. Use the contingency analysis method to validate the Likert scale assessment presented throughout the paper 
  14. The authors also advised to link their results to the previous work shared in the ASHRAE global comfort database II which is available online throughout this link - http://www.comfortdatabase.com/ .
  15. The authors research work is noteworthy contribution the the ASHRAE Global Thermal Comfort Database II - Please donate your dataset to this link - https://datadryad.org/stash/dataset/doi:10.6078/D1F671
  16. Comparing with the result from the ASHRAE global thermal comfort database II with the same region might be interesting 
  17. The air velocities are completely neglected. Whereas they are significant to restore the thermal comfort. Please provide more experiments in the Results section to demonstrate 
  18. In this work, the indoor movement is exempted. In fact, the indoor air movement will have a high impact for reaching neutral thermal sensation. It can make the occupants feel comfortable in relatively high indoor temperature
  19. The other concern is that the reviewer realised that the experiments was being conducted in the climate chamber. The reviewer afraid that this result cannot be generalised for the whole population sample due to the respondent limitation which can generate results. Please provide more evidence on this matter in order to increase the scientific soundness of this paper
  20. There are three thermal adaptation types. They are physiological, which is related to the body reaction due to the temperature change, psychological which is derived from the state of mind of previous experiences and behaviour related adaptation (Brager and de Dear, 1998). Comfort can be reached if there are sufficient opportunities for people to adapt. The comfortable temperature is changeable rather than fixed. (Fergus Nicol and Roaf, 2015). Please consider these theoretical information while you are interpreting the Section 4.1
  21. In order to capture the wider types of occupants and not create the direct generalization which introduce higher bias, the reviewer proposed not to process data as a whole, but to make processing/analysis in clusters. The consideration of using a specific type of air conditioner will be used to decide the cluster of the indoor thermal conditions. This will result in the better mapping of the thermal comfort and occupants' habit in relation of their energy spending. With this cluster groups, analysis can be one within clusters to form more uniform data. If needed, the general conclusion can be drawn using the comparison against clusters. With this approach, the reviewer consider that the result will have less bias if the case is generalized for the whole region. The authors can conduct cluster analysis in statistics to verify the accuracy of their experiments
  22. Relate your conclusions to your research questions

Round 2

Reviewer 3 Report

The authors have been addressed the most of the recommendations satisfactorily therefore I would expect to see the systematic literature review will be conducted by using the visualisation tool. This could help the authors to increase both scientific soundness and credibility of their research work.